# Teaching Robots with Show and Tell: Using Foundation Models to Synthesize Robot Policies from Language and Visual Demonstrations

**Michael Murray**     **Abhishek Gupta**     **Maya Cakmak**
Paul G. Allen School of Computer Science and Engineering
University of Washington
{mmurr, abhgupta, mcakmak}@cs.washington.edu

**Abstract:** We introduce a modular, neuro-symbolic framework for teaching robots new skills through language and visual demonstration. Our approach, SHOWTELL, composes a mixture of foundation models to synthesize robot manipulation programs that are easy to interpret and generalize across a wide range of tasks and environments. SHOWTELL is designed to handle complex demonstrations involving high level logic such as loops and conditionals while being intuitive and natural for end-users. We validate this approach through a series of real-world robot experiments, showing that SHOWTELL out-performs a state-of-the-art baseline based on GPT4-V, on a variety of tasks, and that it is able to generalize to unseen environments and within category objects. Supplementary materials and videos are available on our webpage: https://robo-showtell.github.io

**Keywords:** learning from demonstration, language model planning, neuro-symbolic reasoning

## 1 Introduction

General purpose robots have the potential to enhance productivity and reliability in human-centric, task-oriented settings such as kitchens, warehouses, and offices, but one of the key challenges to achieving this potential is that each environment, user, and task combination demands tailored behavior from the robot. Programming by Demonstration (PbD) is a popular approach to this challenge, enabling end-users to program robots for personalized tasks and environments by providing demonstrations of the desired behavior. However, existing PbD techniques typically require a large number of demonstrations or are unable to extract high-level task information related to control flow and action parameterization from the demonstrations. Natural language is another popular method for personalizing robot behavior, but low-level details can be cumbersome and error prone to communicate with language alone, and most works assume a fixed set of low-level primitives are available to be composed by the language instructions, limiting the scale and complexity of tasks that can be programmed.

Studies in human observational learning show that humans use both language and demonstration when teaching each other new tasks, with both communication modalities playing important roles [1, 2]. Language allows us to transmit abstract information, while demonstrations instantiate that information in concrete examples [3]. Inspired by this insight, we envision a more natural and intuitive PbD system, where end-users can program robots for personalized tasks and environments by flexibly using both language and visual demonstrations.

In this work, we propose SHOWTELL – a system that enables end-users to teach robots new tasks the same way they would teach another person, by visually demonstrating and verbally describing what they are doing as they demonstrate. SHOWTELL is modular, composing a set of pre-trained large language models (LLMs) and vision-language models (VLMs) to synthesize robot policies that

8th Conference on Robot Learning (CoRL 2024), Munich, Germany.

can jointly reason about the language and visual components of the demonstrations and execute the demonstrated behavior in novel scenes. SHOWTELL is designed to be generalizable across a wide range of tasks and environments, and to be intuitive and natural for end-users to use, while requiring only a single demonstration with no additional training or fine-tuning.

While both language and demonstration are traditionally popular interfaces for task planning, there has been comparatively little attention paid to combining the two. Most existing works attempt to train end-to-end models, an approach that is difficult to scale because it requires enough training examples to implicitly learn to understand all demonstrations and perform all tasks within the forward pass of a neural network. In contrast, our approach requires no additional training, utilizing the vast knowledge encoded in off-the-shelf foundation models. Our approach is neuro-symbolic in that it aims to combine the strengths of neural networks, which excel at learning from data, with symbolic reasoning, which excels at manipulating symbols and logical rules, to create a more robust robot PbD system that is modular, allowing us to scale to new tasks by composing existing modules in novel ways, and also also interpretable, allowing us to understand the reasoning behind the generated policies and to easily modify or extend them.

Through real-world robot experiments, we validate our approach, showing that SHOWTELL is able to synthesize robot policies from language and visual demonstrations including tasks that require high level logic such as conditions, iteration, and segmentation. We show that our approach outperforms a state-of-the-art baseline on a variety of tasks, and that it is able to generalize to new objects and environments, while requiring only a single demonstration with no additional training or fine-tuning. We believe that our approach has the potential to significantly improve the usability and performance of robot PbD systems, and to enable robots to learn from demonstrations that are more natural and intuitive for end-users.

## 2   Related Work

**Programming by Demonstration.** Programming by Demonstration, also referred to as Learning from Demonstration or Imitation Learning, has been the subject of four decades of robotics research [4, 5]. Approaches are often categorized based on the method of providing demonstrations and in contrast to methods that require moving the robot (e.g. through teleoperation [6, 7, 8] or kinesthetic teaching [9, 10, 11]), or the use of specialized demonstration hardware [12], in this work we focus on programming by passive observation, where the robot is programmed by observing a human perform the desired behavior [13, 14, 15, 16, 17, 18]. Most existing works attempt to train end-to-end models, an approach that is difficult to scale because it requires enough training examples to implicitly learn to understand all demonstrations and perform all tasks within the forward pass of a neural network. Instead, we propose to leverage the prior knowledge encoded in pre-trained foundation models to synthesize programs that can both reason about demonstrations and execute the demonstrated behavior on a robot with novel scenes and objects.

**LLMs for Task and Motion Planning.** With the advent of large-scale pre-trained language models, there has been growing interest in using these models for robotics tasks. A large body of work has focused on planning and reasoning from text-based natural language instructions [19, 20, 21, 22, 23]. These works typically output their plans as a sequence of robot actions, but recent approaches show the benefits of using LLMs to synthesize code with logical constructs that can be executed by a robot [24, 25, 26]. While much progress has been made in synthesizing code policies from text-based instructions, there has been comparatively little work on synthesizing code policies from visual demonstrations. Wang et al. [27] assume that a demonstration has been converted to a textual description, and focus on generating policies from the text. In contrast, we focus on generating policies directly from visual demonstrations. Most similar to our work, Wake et al. [28] propose to use a VLM to summarize visual demonstrations and generate policies from the resulting summaries. In contrast, we propose to synthesize programs that can reason about the visual demonstration, which allows us to handle more complex demonstrations, better align the language with the visual

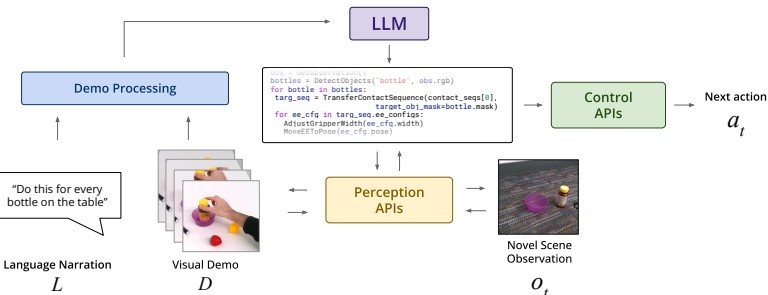

Figure 1: An overview of the SHOWTELL framework. First, the visual and spoken components of the demo are processed and fed into an LLM. The LLM synthesizes a modular program that can jointly reason about the provided demonstration and novel observations to transfer the demonstrated skill to new scenes.

components of the demonstration, and more easily interpret the reasoning behind the generated policies.

**LLMs for Visual Reasoning.** Visual reasoning approaches have typically used end-to-end trained models, but recent works have shown that pre-trained LLMs can be leveraged to accomplish state-of-the-art for many visual reasoning tasks. Early iterations represent the visual information from images as text via captions, objects, and attributes and feed this textual representation to LLMs along with task instructions and in-context examples [29]. Zeng et al. [30] propose a modular approach wherein the LLM leverages other pre-trained models such as vision-language models (VLMs) and audio-language models through program generation. More recent works have scaled this idea to additional tasks including knowledge tagging, image editing, and causal/temporal reasoning over videos [31, 32]. In this work we seek to leverage LLMs for visual reasoning in the context of robot PbD by using the LLM to generate modular programs that jointly reason over video demonstrations and spoken language instructions to understand the task being demonstrated and ground that understanding to robot actions in new scene observations.

## 3 Method

We present SHOWTELL, a neuro-symbolic robot PbD framework for synthesizing modular, generalizable, and interpretable robot manipulation programs from visual demonstrations and natural language. Our approach, illustrated in Figure 1, requires only a single demonstration with no additional training or fine-tuning, as it utilizes a combination of pre-trained foundation models and hand-engineered components to reason about observed demonstrations and to transfer learned skills to new scenes. In the following sub-sections we first formalize the problem setting, then we provide a high-level overview of the approach, and finally we describe each phase of the approach.

### 3.1 Problem Formulation

In this work, we consider multi-modal PbD for robotic manipulation tasks. Let $\mathcal{A}$ be the set of robot actions, and $\mathcal{S}$ the set of world states. We assume access to a human demonstration consisting of visual demonstration component $D = \langle d_0, d_1, \ldots, d_{T_D} \rangle$, where each demonstration frame $d_t$ is an RGB-D image at time $t$, and a spoken language component $L$. Given the demonstration $D$, language $L$, and an initial state $s_0 \in \mathcal{S}$, the goal is to generate an execution $\xi = \langle s_0, a_0, s_1, a_1, \ldots, s_{T_\xi}, a_{T_\xi} \rangle$, where $a_t \in \mathcal{A}$ is an action taken by the robot at time $t$, $s_t \in \mathcal{S}$ is the state before taking $a_t$, and $s_{t+1} = \mathcal{T}(s_t, a_t)$ under environment dynamics $\mathcal{T} : \mathcal{S} \times \mathcal{A} \to \mathcal{S}$. The state $s_t$ is defined by the environment layout, the poses and states of all objects, and the pose and state of the robot. The robot does not directly have access to the state $s_t$, but only to an observation $o_t$. An observation $o_t = (I_t, K_t)$ includes an RGB-D camera image $I_t$ and the robot's proprioceptive state $K_t$. The task

is considered successful if the goal-conditions corresponding to the demonstration $D$ and spoken language $L$ are true at the final state $s_{T_\xi}$.

## 3.2 Overview

Our method utilizes a mixture of pre-trained foundation models and hand-engineered modules to both understand the provided demonstration and to transfer that understanding to new scenes. We first pre-process the demonstration to make relevant information readily accessible for program synthesis. Next, an LLM is used to compose a set of modules into a program. The modules available to the LLM include tools for visual and spatio-temporal reasoning about the video demonstration frames, tools for aligning the spoken language of the demonstration with the visual demonstration, tools for transferring the demonstrated skill to the current environment, and tools for controlling the robot.

## 3.3 Pre-processing

The first step in our approach is to pre-process the demonstration to make relevant information readily accessible to the LLM. The spoken component of the demonstration, $L$, is transcribed using the Whisper speech-to-text model [33]. The transcription is saved and uttered words are indexed by their timestamp relative to the start of the demonstration. The visual component of the demonstration is processed to detect human hands and their interactions with objects in the scene. We utilize the Mediapipe [34] hand landmark detection model, and detect the human hand pose represented as 21 landmarks following the topology in [35]. The two landmarks on the thumb and another two on the index finger are used to represent a parallel jaw robot gripper.

For robot manipulation tasks, timesteps in which the end-effector interacts with objects in the environment are particularly important, so we seek to identify and extract contiguous *contact sequences*, or clusters of timestamps where the hand is in contact with an object, from the demonstration $D$. We extract information about the hands in the scene and the objects that they contact using 100DOH [36], a hand-object interaction model that has been pre-trained on 100K images extracted from a large-scale video dataset of humans interacting with objects. We use 100DOH to extract, for each demonstration frame, a hand bounding box, in-contact object bounding box, and a boolean contact variable indicating whether the hand is in contact with an object or not. We obtain fine grained masks for both hands and objects using Segment Anything Model (SAM) [37] with bounding boxes provided as prompts. 3D perception of the scene is crucial for manipulation tasks, so we additionally produce a point cloud for each demonstration frame using the RGB-D image and camera intrinsics.

## 3.4 Program Synthesis

SHOWTELL uses a powerful code-generating language model (GPT-4 [38]) to synthesize programs that can both reason about a demonstration and execute the demonstrated behavior in a new scene. The LLM is provided with a prompt that consists of import statements and API documentation that specifies the available functions, a brief summary of contact sequences extracted during the pre-processing phase, and the full transcribed narration. The program is run with the Python interpreter, allowing the use of control flow tools like for loops, conditionals like if/else, built in functions like sort, and built in modules such as datetime or math. All of the

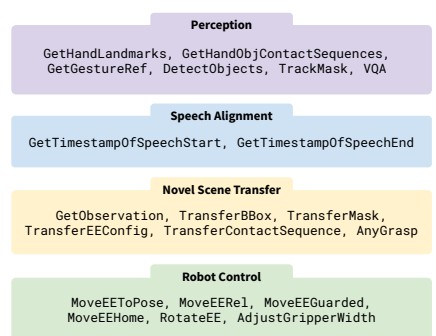

Figure 2: Taxonomy of modules available to the synthesized neuro-symbolic programs.

information extracted during the pre-processing phase is made available in addition to a suite of modules. The additional modules made available to the program include a mixture of pre-trained

foundation models and hand-engineered modules, including modules for perception, speech alignment, transferring to novel scenes, and controlling the robot. In the following sections we describe each of the custom modules and their APIs and a taxonomy of the modules is provided in Figure 2.

### 3.4.1 Perception

The perception modules made available to the neuro-symbolic program enable extracting information about both the visual demonstration, $D$, and the new scene observations, $\langle o_0, o_1, \ldots, o_{T_\xi} \rangle$. The perception modules include APIs for accessing visual information made available in the pre-processing step via `GetHandObjectContactSequences`. The perception modules also include APIs for free-form visual reasoning, `DetectObjects`, `TrackMask`, and `VQA`. The `DetectObjects` module uses ViLD [39], a pre-trained open-world object masking VLM. Detected masks can be tracked to new frames and scenes with the `TrackMask` module, which uses the XMem mask tracking model under the hood [40]. The `VQA` module is a pre-trained visual question answering VLM, BLIP-v2 [41], that can be used to answer questions about the visual scene.

### 3.4.2 Language Alignment

To understand a multi-modal demonstration, the robot must align the spoken language of the demonstration with the visual demonstration. For this purpose, we provide the LLM with two modules, `GetTimestampOfSpeechStart` and `GetTimestampOfSpeechEnd`, which are used to extract the start and end timestamps of spoken words from the language component of the demonstration, $L$, and enable the program to align the language with timestamps corresponding to frames in the visual demonstration, $D$.

### 3.4.3 Novel Scene Transfer

Ultimately, the goal is to transfer the policy generated from the demonstration to new environments. To do this, the robot must match the visual demonstration $D$ to the current environment observed in $o_t$. The `TransferBBox`, `TransferMask`, `TransferEEConfig`, `TransferContactSequence` modules are used to transfer detections to new scene observations. These modules find corresponding reference points by leveraging features from Stable Diffusion [42], a pre-trained image diffusion model which have been shown to implicitly encode rich information about the structure of objects within an image, and have been shown to be highly effective for finding corresponding points for visual reasoning tasks [43, 44, 45]. Using these features, we can match similar points on within-category objects in addition to exact points. For example, after demonstrating how to pick up a mug by its handle, the robot should be able to repeat this skill for visually distinct mugs and mugs of different sizes.

In order to facilitate grasping of objects in the new scene, we provide the `AnyGrasp` module, which is used to find the best grasp configuration for a particular object mask in the scene. This module uses a pre-trained grasp prediction model [46] to predict the best grasp configuration for a particular object based on its point cloud, and can be used to find grasp configurations for objects that were not present in the demonstration.

### 3.4.4 Robot Control

To control the robot we provide a set of modules to specify end-effector goal poses and configurations. The modules `MoveEEToPose` is used to specify a goal EE pose in 3D space, `MoveEERel` is used to specify a goal relative to the current EE pose, and `MoveEEGuarded` performs a guarded movement along a given vector until contact occurs. The module `RotateEE` is used to specify a goal EE rotation. And finally, `AdjustGripperWidth` is used to specify a desired gripper width to open and close the gripper. These modules are used to specify the robot's end-effector goal poses and configurations, which are used to generate the robot's actions during program execution as described in Section 3.5.

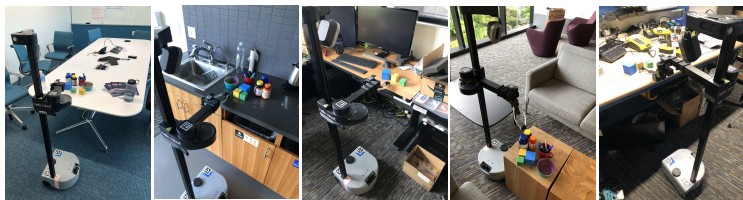

Figure 3: We evaluate SHOWTELL using a Stretch RE2 robot to perform 16 real world manipulation tasks across 5 visually distinct environments including a conference room, a kitchen, a classroom, an office lounge, and a cluttered workbench. We show that this approach is effective for teaching manipulation tasks requiring high level logic such as conditions, iteration, and segmentation.

## 3.5   Skill Execution

After program synthesis, the resulting program is run with a Python interpreter and its execution is a simple Python call. During execution, correspondence matching modules are evaluated to transfer reference points identified in the demonstration to the new scene observation, and all waypoints are redefined relative to the reference points in the new scene. To interpolate robot motion between end-effector goal poses, we use a collision-free motion planner to generate a trajectory of robot actions for reaching the next desired waypoint goal. Specifically, we use the GPU accelerated motion generation library cuRobo [47].

## 4   Experimental Setup

**Hardware and Environments:** To evaluate our approach, we conduct a series of real world experiments with a Stretch RE2 robot [48] across 5 indoor environments as illustrated in Figure 3. The robot's mobile base, arm lift, and telescoping arm are moved in conjunction to reach 6-DOF target waypoints. The robot's end effector is a parallel-jaw gripper with rubber fingertips. An Intel RealSense D435i RGB-D camera is mounted to the frame which is used both to record demonstrations and to provide observations during execution.

**Baselines and Experiments:** We evaluate our approach against GPT4-V-Robot [28], a state-of-the-art method for robot task planning from language and visual demonstrations using LLMs. In this baseline approach, demonstration frames are directly fed to a VLM (GPT4-V [38]), which generates a summary that is used by the LLM to generate a policy. To study the contribution of the visual and spoken components of the demonstration we perform a series of ablation experiments. First we evaluate a language-only baseline, ShowTell-NoVis, with no access to the visual demonstration, similar to the approach proposed by Liang et al. [24]. We also include a vision-only baseline, ShowTell-NoLang, which uses the visual demonstration only to generate policies without access to the language component.

**Evaluation Tasks:** We evaluate SHOWTELL using a set of 16 real world manipulation tasks across 5 visually distinct environments. We group evaluation tasks based on the challenge conditions they present. First we evaluate with simple demonstrations that are straightforward to follow, including pick and place, stacking cubes, and opening drawers. Next we evaluate tasks involving control flow including iterative tasks, for example *"move all of the blocks over to this container"*, and tasks with conditionals, for example *"open the drawer only if it is closed"*. Finally, we evaluate tasks that require segmenting the demonstration, for example if the demonstrator makes an error and corrects themselves, the robot is required to segment the demonstration into correct and incorrect segments to generate a policy that correctly imitates only the desired behavior. All of the tasks, including the simple tasks, can require the robot to jointly reason about the visual and spoken components of the demonstration to resolve ambiguities. For example, an instruction like *"pick up this mug and move it over here"* requires aligning the spoken instruction with the visual demonstration to determine which mug to pick up and where to move it.

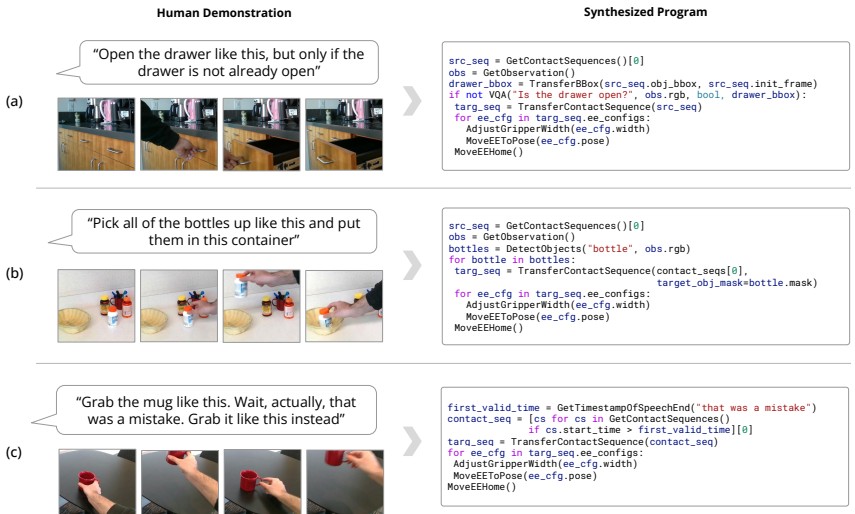

Figure 4: Qualitative examples of code synthesized by SHOWTELL for a set of representative demonstrations. The representative demonstrations show the ability to follow high level logic including (a) conditionals (b) iteration and (c) segmentation.

| Demo Type | ShowTell-NoLang | | ShowTell-NoVis | | GPT4-V-Robot [28] | | ShowTell (ours) | |
|---|---|---|---|---|---|---|---|---|
| | GCR | SR | GCR | SR | GCR | SR | GCR | SR |
| Simple | 0.89 | 0.85 | 0.86 | 0.81 | 0.88 | 0.85 | 0.96 | 0.94 |
| Iterative | 0.31 | 0.12 | 0.61 | 0.59 | 0.43 | 0.22 | 0.94 | 0.85 |
| Conditional | 0.41 | 0.39 | 0.64 | 0.64 | 0.41 | 0.41 | 0.93 | 0.93 |
| Segmented | 0.12 | 0.06 | 0.48 | 0.42 | 0.20 | 0.18 | 0.93 | 0.91 |

Table 1: Real-world robot manipulation task performance across different task families.

**Metrics:** We use two primary metrics to evaluate system performnace: *Success Rate (SR)* and *Goal Condition Recall (GCR)*. The task-relevant goal conditions are the set of state changes that must be satisfied at the end of an episode for the task to be considered successful. *SR* is the fraction of rollouts for which the object positions and state changes completely satisfy the task goal-conditions at the end of the action sequence. *GCR* is the fraction of goal-conditions successfully completed at the end of an episode to those necessary to have finished a task.

# 5  Results

In our experiments we seek to answer the following research questions: 1) Is SHOWTELL practical for teaching a range of robot manipulation skills including those that require high level logic like conditions, iteration, and segmentation? 2) How does SHOWTELL compare to existing state-of-the-art methods for generating robot policies from language and visual demonstrations? 3) Is SHOWTELL able to generalize to unseen environments and within category objects? 4) To what extent do the visual and spoken components of the demonstration contribute to the performance?

To evaluate the performance of SHOWTELL, we perform a total of 50 rollout trials per task: 10 demonstrations are provided (2 demonstrations in each environment) and 5 rollouts are performed for each demonstration, resulting in a total of 50 rollouts per task. We group tasks based on challenge conditions as described in section 4 and for each task family, we report the average SR and GCR across all rollouts.

The results summarized in Table 1 show that SHOWTELL outperforms existing state-of-the-art methods for generating robot policies from language and visual demonstrations. Even for

simple tasks, SHOWTELL outperforms the baseline methods, as the modular, neuro-symbolic programs are more easily able to reason about the demonstration and to resolve ambiguities than the monolothic VLM used to summarize demonstrations in the baseline method. This advantage is even more pronounced for the other task families as the purely sequential structure of the baseline method is not well suited to tasks that require high level logic like conditions, iteration, and segmentation. The qualitative examples in Figure 4 illustrate the ability of SHOWTELL to synthesize programs from such demonstrations. Comparing the ablation experiments, we see that the visual and spoken components of the demonstration are both important for generating effective policies, as the ablation experiments ShowTell-NoVis and ShowTell-NoLang perform poorly compared to the full SHOWTELL approach, even for simple tasks. This is be-

| Rollout | SR |
|---|---|
| Canonical | 0.92 |
| Unseen environment | 0.84 |
| Unseen objects | 0.88 |

Table 2: Performance across rollouts: in the canonical scene, with unseen objects, and in unseen environment.

cause the visual and spoken components of the demonstration are often complementary, and help to resolve ambiguities present in the individual components of the demonstration. To better understand the failure cases, we analyze the failures in Section C of the appendix and illustrate the distribution of failures in Figure 7. We finally evaluate the performance of SHOWTELL across rollouts in the canonical scene (the scene used for demonstration), with unseen objects, and in unseen environments. The results summarized in Table 2 indicate that SHOWTELL is able to generalize to unseen environments and within category objects, while requiring only a single demonstration with no additional training or fine-tuning.

## 6 Limitations

Our approach has several limitations. Most crucially, the approach is constrained by the limitations of the pre-trained models it uses, although we note that the modularity of the framework allows for easy integration of new models as they become available. As parallel fields progress and new models are developed, we expect that the performance of SHOWTELL will improve and the variety of applicable skills will expand. Additionally, fine-tuning of pre-trained models to mitigate this limitation and reduce the need for explicit pre-processing is an exciting direction for future work. Our approach is also limited by the quality of the demonstrations provided, assumes demonstrated grasps can be mapped to a robot gripper, and may struggle with demonstrations that are incomplete or difficult to perceive due to occlusion or poor quality. Future work could leverage the interpretability of the approach to mitigate this limitation through interactive program repair. The inclusion of both visual and language inputs may introduce ambiguities that could complicate the approach. Future works could incorporate interactive dialog with the user to disambiguate inputs that are ambiguous or unclear. Finally, the approach uses closed loop execution to reach each goal waypoint and may struggle with dynamic disturbances or changes in the environment during execution.

## 7 Conclusion

In this work, we present SHOWTELL, a neuro-symbolic robot PbD framework for synthesizing modular, generalizable, and interpretable robot manipulation programs from visual demonstrations and natural language. Our approach can teach robot manipulation skills from a single demonstration, without requiring any additional training or fine-tuning, by utilizing a combination of pre-trained foundation models and hand-engineered components to reason about observed demonstrations and to transfer learned skills to new scenes. We evaluate SHOWTELL on a set of 16 real world manipulation tasks across 5 visually distinct environments and show that this approach is effective for teaching a wide range of manipulation tasks. We show that SHOWTELL is able to reason about demonstrations that require high level logic like conditions, iteration, and segmentation, and that it outperforms existing state-of-the-art methods for generating robot policies from language and visual demonstrations.

**Acknowledgments**

This research is partially funded by the UW + Amazon Science Hub. We would like to thank Entong Su for help with cuRobo, Varad Dhat for robot hardware maintenance, Sylvia Dai, Michael Wolf, Joshua Smith, Markus Grotz, and Nick Walker for helpful discussions.

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

# Appendix

## A  Implementation Details

### A.1  Perception APIs

The perception APIs provide access to the visual information made available in the pre-processing step, in addition to open world perception of both the demonstration scene and the novel scene via pre-trained foundation models. The APIs include:

- `GetContactSequences()` Returns all of the hand-object contact sequences detected in the demo by 100DOH [36] during the preprocessing step.

- `GetHandLandmarks(timestamp)` Returns the thumb and index finger hand landmarks detected in the demo by Mediapipe [34] during the preprocessing step.

- `GestGestureRef(timestamp)` Returns a mask of the object being referred to at the given timestamp of the demo. Any in-hand object is used, otherwise a vector is produced from the hand landmarks and the closest object to the vector is used.

- `DetectObjects(obj_class, rgb_frame)` Returns a mask of the object class in the given RGB frame using ViLD [39].

- `TrackMask(mask, origin_frame, new_frame)` Tracks the given mask to the given RGB frame using XMem [40].

- `VQA(query, rgb_frame, output_type, bbox)` Returns the answer to the given query about the given RGB frame using BLIP-v2 [41].

### A.2  Speech Alignment APIs

The speech alignment APIs provide access to the spoken language of the demonstration and the timestamps of the spoken words. The spoken language is transcribed using the Whisper speech-to-text model [33]. The transcription is saved and uttered words are indexed by their timestamp relative to the start of the demonstration. The APIs include:

- `GetTimestampOfSpeechStart(speech, occurance)` Returns the timestamp of the start of the given occurance of the given speech in the spoken language of the demo.

- `GetTimestampOfSpeechEnd(speech, occurance)` Returns the timestamp of the end of the given occurance of the given speech in the spoken language of the demo.

### A.3  Scene Transfer APIs

The scene transfer APIs provide access to the tools necessary to transfer the demonstrated skill to the current environment observed in the new scene. The APIs include:

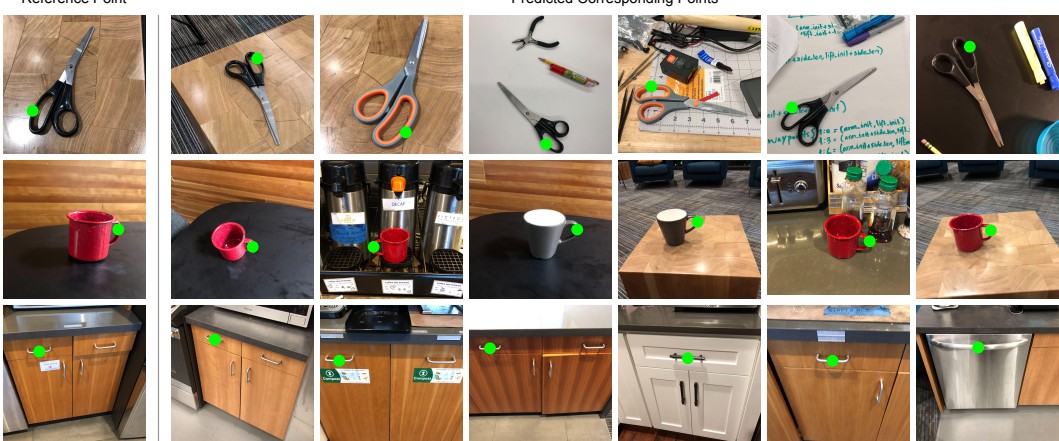

Reference Point        Predicted Corresponding Points

Figure 5: During scene transfer, corresponding reference points are found in a new observation using features from a pre-trained image diffusion model. On the left, reference points are shown in the demonstration scene. On the right, corresponding reference points are shown as detected in novel viewpoints, scenes, and with novel within category objects.

- `TransferContactSequence(contact_seq, target_mask)` Transfers the given contact sequence to the new scene. Points corresponding to each reference point in the contact sequence are identified in the new scene by using features from a pre-trained image diffusion model. This process can be performed over the entire scene or optionally guided by semantic/instance masks.

- `TransferEEConfig(timestamp, reference_point, mask)` Transfers the end-effector configuration from the given timestamp to the new scene using the given reference point as an anchor. A corresponding reference point in the contact sequence is identified in the new scene by using features from a pre-trained image diffusion model. This process can be performed over the entire scene or optionally guided by semantic/instance masks.

- `TransferBBox(bbox, frame)` Transfers the given bounding box to the given RGB frame by using SAM [37] to acquire a mask from the bounding box and XMem [40] to transfer the mask to the new frame before converting back to a bounding box.

- `AnyGrasp(mask)` Returns the best grasp configuration for the given object mask using AnyGrasp [46].

## A.4 Robot Control APIs

The robot control APIs provide access to the tools necessary to specify end-effector goal poses and configurations. All motion is generated using the motion planner implemented by cuRobo [47]. The APIs include:

- `MoveEEToPose(pose)` Moves the end-effector to the given pose in 3D space.

- `MoveEERel(rel_pose)` Moves the end-effector to the given pose relative to the current end-effector pose.

- `MoveEEGuarded(direction)` Moves the end-effector along the given direction until contact occurs.

- `MoveEEHome()` Moves the end-effector to the home position.

- `RotateEE(rotation)` Rotates the end-effector to the given rotation.

- `AdjustGripperWidth(width)` Adjusts the gripper width to the given width.

# B Experimental Details

## B.1 Benchmark Tasks

We design our evaluation tasks to cover a wide range of contact-rich manipulation behaviors involving prehensile and non-prehensile motions. The tasks range from rearranging objects, to multi-step extraction from cluttered scenes, to tool use, to manipulation of deformable and articulated objects. The tasks are grouped based on reasoning challenge conditions they present, including simple tasks, iterative tasks, conditional tasks, and segmented tasks. This taxonomy can be seen in Table 3. Below we provide a brief description of each task and the conditions used to determine successful completion of the task.

- *Pick-and-place*: In this task, the robot picks up a bottle by its top and places it into a bowl. The task is successful if the robot grasps from the top of the bottle and the bottle is contained inside of the bowl at the end of execution.

- *Open drawer*: In this task, the robot is required to open a drawer. This requires a precise grasp of the drawer handle and careful imitation of the demonstrated trajectory to open the drawer. The task is successful if the drawer is open at the end of execution.

- *Stack blocks*: This task demonstrates a manipulation program with a multi-step horizon. The robot must stack a set of three colored blocks in the same order as the demonstration. The task is successful when the blocks are stacked in a stable column following the order given by the demonstration.

- *Fold towel*: This task demonstrates a manipulation program with deformable objects. The robot must first grasp the corner of a towel, then follow the demonstrated trajectory to fold the towel. The task is successful when the towel is folded.

- *Wipe whiteboard*: This task demonstrates a manipulation program with tool use. The robot must first grasp a cloth, then follow the demonstrated trajectory to clean a marking off of a whiteboard using the cloth. The task is successful when the whiteboard is cleaned.

- *Unplug charger*: In this task the robot must first grasp a charger, then follow the demonstrated trajectory to unplug the charger from a wall outlet. The task is successful when the charger is unplugged.

- *Move all bottles to container*: This task demonstrates an interative manipulation program with variable number of objects. For each bottle in the scene, the robot must grasp the bottle and move to the receptacle as demonstrated. The task is successful when all bottles are contained inside of the container.

- *Clear table*: This task demonstrates an iterative manipulation program with variable number of objects. For each object on the table, the robot must grasp the object and move it to the receptacle as demonstrated. The task is successful when the table is clear.

- *Sort items*: This task demonstrates an iterative manipulation program with variable number of objects and conditional logic. The robot must sort the objects on the table into two groups based on their color. The task is successful when all objects are sorted correctly.

- *Mechanical search*: This task demonstrates an iterative manipulation program with variable number of objects and conditional logic. The robot must search for a target object among a set of distractor objects. The task is successful when the target object is found.

- *Unstack to grasp*: This task demonstrates an iterative manipulation program with variable number of objects and conditional logic. The robot must unstack a set of blocks to reveal a target object, then grasp the target object. The task is successful when the target object is grasped.

- *Conditional pick and place*: This task demonstrates a manipulation program with conditional logic. The robot must move an object to a receptacle only if the receptacle is empty. The task is successful when the object is contained inside of the receptacle.

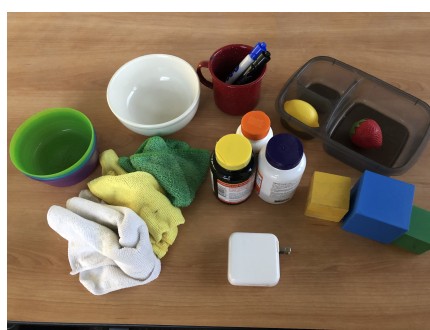

Figure 6: Benchmark items used in the evaluation of SHOWTELL.

- *Open drawer if not opened*: This task demonstrates a manipulation program with conditional logic. The robot must open a drawer only if it is closed. The task is successful when the drawer is open.

- *Conditional grasp*: This task demonstrates a manipulation program with conditional logic. The robot must grasp an object in one of two ways based on the object's color. The task is successful when the object is grasped correctly.

- *Assemble bento*: This task demonstrates a manipulation program with segmentation. The robot must assemble a bento box by placing each item in the correct compartment. The task is successful when the bento box is assembled correctly.

- *Fixed grab mug*: This task demonstrates a manipulation program with segmentation. The robot must grasp a mug by its handle. The task is successful when the mug is grasped correctly.

## B.2 Evaluation Details

We evaluate the performance of SHOWTELL with a total of 50 rollout trials per task: 10 demonstrations are provided (2 demonstrations in each environment) and 5 rollouts are performed for each demonstration, resulting in a total of 50 rollouts per task. For tasks that require iteration we vary the number of objects in the scene for each demonstration. For tasks that require conditionals we vary the state of the environment for each demonstration. To evaluate the performance of SHOWTELL across rollouts with unseen objects we perform an additional 10 rollouts each for the *pick-and-place*, *clear table*, and *conditional grasp* tasks with objects that were not present in the demonstration. To evaluate the performance of SHOWTELL across rollouts in unseen environments we perform an additional 10 rollouts each for the *pick-and-place* and *open drawer* tasks in a distinct environment from the demonstration.

| Task Name | Simple | Iterative | Conditional | Segmented |
|---|:---:|:---:|:---:|:---:|
| Pick and place | ✓ | | | |
| Open drawer | ✓ | | | |
| Stack blocks | ✓ | | | |
| Fold Towel | ✓ | | | |
| Wipe whiteboard | ✓ | | | |
| Unplug charger | ✓ | | | |
| Move all bottles to container | | ✓ | | |
| Clear table | | ✓ | | |
| Sort items | | ✓ | | ✓ |
| Mechanical search | | ✓ | ✓ | |
| Unstack to grasp | | ✓ | ✓ | |
| Conditional pick and place | | | ✓ | |
| Open drawer if not opened | | | ✓ | |
| Conditional grasp | | | ✓ | ✓ |
| Assemble bento | | | | ✓ |
| Fixed grab mug | | | | ✓ |

Table 3: Real-world robot manipulation tasks across different task families.

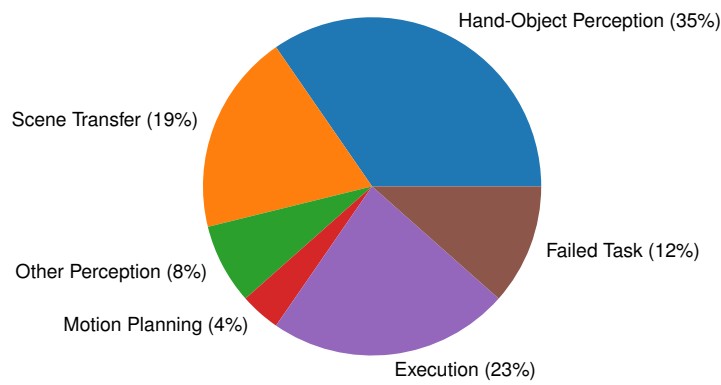

Figure 7: The distribution of failures across SHOWTELL benchmark tasks. Executions can fail due to errors in hand-object perception, errors in scene transfer correspondence matching, other perception failures, failure to motion plan, control execution failures, or failure to meet the task requirements.

## C    Failure Analysis

We analyze the failures of SHOWTELL across the benchmark tasks. The distribution of failures across the tasks is shown in Figure 7. We divide failures into six categories:

- **Hand-Object Perception:** Errors in hand-object perception, where the robot fails to correctly identify the hand or hand-object contact.

- **Scene Transfer Correspondence Matching:** Errors in scene transfer correspondence matching, where the robot fails to correctly match reference points in the demonstration to the new scene.

- **Other Perception Failures:** Errors in object detection, tracking, or other perception tasks.

- **Failure to Motion Plan:** Errors in motion planning, where the robot fails to plan a collision-free trajectory.

- **Control Execution Failures:** Errors in control execution, where a motion plan is generated but the robot fails to execute the plan correctly.

- **Failure to Meet Task Requirements:** Errors in the manipulation program, incorrect execution, or other factors that prevent the robot from successfully completing the task.

The most common failure mode is due to errors in hand-object perception, indicating that improvements in hand-object perception could lead to significant improvements in overall performance. Execution is another common source of failure, typically due to imprecise control of the robot's wheeled base. More dynamic control involving visual servoing could mitigate this failure mode in future work. The other failure modes are less common, but still contribute to overall performance.

```
You are writing Python code to control a robot based on a natural language instruction and a visual demonstration. The
natural language instruction is provided in the comments and you can query information about the visual demonstration.
You must write code to understand what is being demonstrated in the input video and execute the demonstrated behavior
with the robot. Please output only valid Python code with no other explanations. Do not output markdown or comments,
just the executable python code.
```

**User Message:**

```
The APIs available are:
- GetContactSequences()
  Description: Returns all of the hand-object contact sequences detected in the demo. Each contact sequence has (start_time,
end_time, ref_frame, ee_configs)
- GetHandLandmarks(timestamp)
  Description: Returns the hand landmarks detected at the given timestamp, one (u,v) point for the index finger and one for the
thumb.
- GetObservation()
  Description: Returns the current observation frame from the robot's camera, one RGB array and one depth.
- GetGestureRef(timestamp)
  Description: Returns a mask of the object being referenced at the given timestamp, either in-hand or being gestured to.
- DetectObjects(obj_class, rgb_frame)
  Description: An open-world object detector that returns a mask and bounding box of the object detected in rgb_frame
- TrackMask(mask, origin_frame, new_frame)
  Description: track a mask from one frame to another frame
- VQA(query, rgb_frame, output_type, bbox=None)
  Description: answers a visual question about the rgb_frame. output_type can be bool, number, or string. bbox is optional.
- GetTimestampOfSpeechStart(speech, occurence_idx)
  Description: returns the timestamp when the given speech started for the occurence_idx time
- GetTimestampOfSpeechEnd(speech, occurence_idx)
   Description: returns the timestamp when the given speech ended for the occurence_idx time
- TransferContactSequence(contact_sequence, target_obj_mask=None)
  Description: Transfer the given contact sequence to the new scene. Optionally provide a mask of the target object in the new scene.
- TransferBBox(bbox, frame)
   Description: Transfer the bounding box in the given frame to the new scene observation.
- TransferEEConfig(timestamp, reference_point, mask=None)
   Description: Transfer the end effector pose at the given timestamp to the new scene observation.
- AnyGrasp(point_cloud)
   Description: Chooses a grasp configuration for an arbitrary point cloud
- MoveEEToPose(pose)
  Description: Move the EE to the specified goal pose
- MoveEERel(delta_pose)
  Description: Move the EE to the specified relative pose
- MoveEEGuarded(vector)
  Description: Move EE along a given vector until contact
- MoveEEHome()
  Description: Return EE to home position. Usually done at the end of the task.
- AdjustGripperWidth(width)
   Description: Adjust the gripper width (used to grasp or ungrasp)

You can use color_frames[timestamp] to access the RGB frame for any timestamp in the demo video
You can use depth_frames[timestamp] to access the depth frame for any timestamp in the demo video
You can use pointclouds[timestamp] to access the point cloud for any timestamp in the demo video
You can use obs = GetObservation() to get the current observation frame from the robot's camera

Do not output markdown (e.g. "```python") or add any code comments, just the executable python code.

For example, if the recorded demo has the following contact sequences:
(1) PICK_AND_PLACE, target_object="pill bottle", target_receptacle="bowl"

And the natural language narration of the demo is as follows:
Grasp the pill bottle and move it to the bowl.

You can write code to execute the demonstrated behavior with the robot similar to this:
src_seq = GetContactSequences()[0]
obs = GetObservation()
targ_seq = TransferContactSequence(src_seq)
for ee_cfg in targ_seq.ee_configs:
    AdjustGripperWidth(ee_cfg.width)
    MoveEEToPose(ee_cfg.pose)
MoveEEHome()

The recorded demo has the following contact sequences
{contact_sequence_summary}

The natural language narration of the demo is as follows:
{demo_narration}
```

Figure 8: The prompt includes API documentation that specifies the available functions, a brief summary of contact sequences extracted during the pre-processing phase, and the full transcribed narration of the demonstration.

