# OpenReview forum: "Teaching Robots with Show and Tell: Using Foundation Models to Synthesize Robot Policies from Language and Visual Demonstration"
_robot-learning.org/CoRL/2024/Conference — CoRL 2024_

### Official Review · Reviewer_2euh · 2024-07-19
**Interesting Approach and Great Results**

**Originality:** 4
**Technical Quality:** 5
**Clarity Of Presentation:** 4
**Potential Impact:** 3
**Recommendation:** 3
**Confidence:** 5

**Review:**

The paper presents a well-structured and methodologically sound approach to teaching robots through the SHOWTELL framework. The experiments conducted are thorough, involving real-world scenarios that validate the effectiveness of the proposed method. The writing is generally clear and accessible, with a logical flow that guides the reader through the problem statement, methodology, and results. The integration of language and visual inputs in a modular neuro-symbolic framework to perform robotic manipulation tasks without any fine-tuning or additional training is a novel and significant contribution.

**Strengths**
1. The framework demonstrates the capability to generalize to new objects and environments without requiring any fine-tuning or additional training, relying solely on a single demonstration.
2. Comprehensive real-world experiments offer compelling evidence supporting the effectiveness of the SHOWTELL framework.
3. The modular architecture facilitates the seamless integration of new pre-trained models as they become available, allowing for straightforward updates to leverage state-of-the-art foundation models.
4. The comparison with strong baseline models yields promising results, demonstrating improvements over existing approaches such as GPT4-V-Robot.



**Weaknesses**

1. The framework may face challenges when dealing with incomplete or poorly executed demonstrations, potentially restricting its effectiveness in real-world applications.
2. The reliance on both visual and language inputs may introduce ambiguities that could complicate the program synthesis if not properly managed. Additionally, there is no mechanism to disregard one of the less reliable or confusing inputs in situations where both inputs are contradictory or unclear.
3. The paper does not discuss or present any failure cases or provide insights on how to resolve them, which limits the understanding of the framework's limitations and potential areas for improvement.
4. The proposed method is closed-loop, which will be a limitation in real-world deployment with dynamic changes and noise.
5. The tasks in the experiments remain short-horizon tasks rather than long, complex tasks that involve conditional multi-step instructions, such as: "If the drawer is open, then retrieve the pen inside it; otherwise, bring me the pen that is on the table."

**Quality Of The Limitations Section:**

1

**Questions For Rebuttal:**

1. Curious how the model performs in unseen environments with unseen objects (both).
2. What would be the implications of fine-tuning certain modules within the framework for handling more complex tasks? Will that reduce the reliance on hand-engineered inputs and the need for pre-processing demonstrations to some degree?
3. Can you provide examples of failure cases encountered during experiments, and what insights have been gained regarding the limitations of the approach and potential solutions?
4. How does the framework adapt to dynamic disturbances or changes in the environment during execution, and what measures are taken to ensure reliable performance under such conditions?
5. Can this framework be adapted to handle long-horizon tasks that involve multiple sub-goals, such as: "If the drawer is open, then retrieve the pen inside it; otherwise, bring me the pen that is on the table"?
6. Is this framework capable of extending to utilising multiple video demonstrations as a sequence of demonstrations for a long horizon task like - Bring me a glass of water and throw this paper in the bin where first part of instruction is shown in one video and the second in another. If we are able to do this then we can compose very complicated tasks by combining short videos for demonstration increase or data as a whole.
7. Is the framework capable of leveraging multiple video demonstrations as a sequence for executing a long-horizon task, such as: "Bring me a glass of water and throw this paper in the bin," where the first part of the instruction is shown in one video and the second in another? If so, how could this capability enable the composition of more complex tasks by combining shorter video demonstrations?

**Robotics Focus:**

4

**Summary Of Paper:**

The paper introduces SHOWTELL, a neuro-symbolic framework that enables robots to learn manipulation skills from a single visual demonstration and natural language instruction, without requiring additional training. It emphasizes generalizability across various tasks and environments, allowing robots to adapt to new objects and settings effectively. SHOWTELL's modular design enhances interpretability and scalability, enabling it to handle complex tasks that involve high-level reasoning. Empirical results demonstrate that SHOWTELL outperforms existing state-of-the-art methods, highlighting its potential to improve the usability and performance of robot programming by demonstration systems.

**Summary Of Recommendation:**

In summary, I appreciate the contributions and innovative approach presented in this paper. The results are convincing and demonstrate the effectiveness of the proposed framework. It appears to be a strong fit for the conference, and I believe it should be accepted. Additionally, it would be valuable to explore some failure cases and discuss potential strategies for addressing them, as this could further enhance the robustness of the approach.

---

### Official Review · Reviewer_F4Hd · 2024-07-20

**Originality:** 3
**Technical Quality:** 4
**Clarity Of Presentation:** 4
**Potential Impact:** 3
**Recommendation:** 3
**Confidence:** 4

**Review:**

The paper is well-written and clearly specifies the problem at hand, why it is important, how it is addressed, and lists a set of interesting questions in the experiments section.

Strenghts:
- The method tackles a non trivial problem of teaching robots via multimodal demonstrations in the form of language and video. The authors study the effect of each of these modalities and what improvement they can bring to robot learning.
- The method seems to be effective in transferring the behaviour to new scenes, and a series of interesting real-world experiments are conducted.

Weaknesses:
- The method relies on a very large set of pre-defined models, primitives, and hand-defined APIs. While they allow the method to work and are well orchestrated by GPT-4, this can become a bottleneck to generalise even more to more complex movements and behavior that go beyond pick and place or relatively linear movements.
- A plethora of recent works have used this formulation of a large VLM + code/off-the-shelf models/motion primitives to tackle similar problems, making it difficult to clearly understand the impact and novelty that this method can bring. Nevertheless, I do know that many papers are concurrent work and it would be unfair to assume the authors can both keep track of them all, especially considering the time between submissions, reviews, etc.
Minor point: I do not see any real benefit, scientifically, in using speech as an input with respect to written language. While I appreciate the authors ability to use it as an input modality, which is more natural than writing, it is becoming more and more a slight engineering challenge instead of a fundamental research questions. I would therefore advocate for simplicity instead of over-engineering, as the paper is already very engineering/system heavy.

Overall, I believe this is an interesting project and an interesting contribution, and a certainly challenging method to develop, however it is difficult to distinguish clear scientific contribution with respect building a system with existing tools (similarly to papers like Dobb-E [1] and OK-Robot [2]: while I know they differ in the kind of inputs, the general concept of building a system with many off-the-shelf models is similar). The experimental section is well executed, and the method is certainly valid. The lack of videos, websites, or supplementary material however makes it harder to more deeply understand the nuances of the method, or see the actual inputs and outputs.

[1] Nur Muhammad "Mahi" Shafiullah*, Anant Rai*, Haritheja Etukuru, Yiqian Liu, Ishan Misra, Soumith Chintala, Lerrel Pinto, Dobb-E: On Bringing Robots Home
[2] Peiqi Liu, Yaswanth Orru, Jay Vakil, Chris Paxton, Nur Muhammad Mahi Shafiullah, Lerrel Pinto, OK-Robot: What Really Matters in Integrating Open-Knowledge Models for Robotics

**Quality Of The Limitations Section:**

3

**Questions For Rebuttal:**

As stated in the review, I would like the reviewers to better clarify the scientific impact of the paper with respect to its engineering impact, also considering if possible the existing papers I mentioned.

**Robotics Focus:**

4

**Summary Of Paper:**

The paper proposes ShowTell, a method to perform Programming by Demonstration by receiving instructions both in the form of language and video. Through a large series of off-the-shelf models, predefined primitives, and APIs, all coordinates by a large VLM, the method can ingest this multimodal data and generate the desired behaviour in new settings-

**Summary Of Recommendation:**

I suggest a weak accept, but I will take into account opinions from other reviewers and make the final decision after the discussion session. To recap, I believe the paper is interesting and the experiments and method are well developed and executed, however it is difficult to really understand what novelty it can present in the wide range of VLM-based papers, and the engineering aspects can surpass the scientific ones, making it look more like a system paper.

---

### Official Review · Reviewer_sbTj · 2024-07-23
**Good paper with limited experiments.**

**Originality:** 3
**Technical Quality:** 3
**Clarity Of Presentation:** 3
**Potential Impact:** 3
**Recommendation:** 3
**Confidence:** 4

**Review:**

## Strengths
- The paper is solid and well-written.
- It addresses the challenging problem of learning control policies for complex tasks, such as "moving all bottles in a tray," from video demonstrations and offers a reasonable solution using vision-language models (VLMs).
- The proposed pipeline significantly outperforms the baseline, with approximately double the performance on iterative and conditional tasks.

## Weakness
The following are the concerns in decreasing order of importance:

- The paper does not include the prompts used to obtain the plan. Since the method of prompting and the choice of in-context examples can significantly impact performance, this omission poses a challenge to the reproducibility of the results.

- While the authors discuss the implementation of some important APIs, details of others are missing (especially the scene transfer APIs). To understand the applicability and limitations of the work, it is important to discuss the implementation of all the APIs (at least in the appendix).

- It appears that the robot is controlled using end-effector poses. However, the process by which waypoints from the demonstrations are transferred to the end-effector positions in the current scene is not clear. Furthermore, the hand-engineered transfer functions may fail if the object pick-and-place distances vary significantly from those seen in the demonstration. This raises concerns about the generalizability of the learned policy.

- The preprocessing seems to assume the human uses two fingers to manipulate the object, which is the case in all examples and figures provided in the paper. This makes the preprocessing and perception API tailored for such videos and may affect performance if the assumptions are violated.

- The authors state that the evaluation is done on 16 tasks. However, the details of the tasks are missing. Additional information on the type of tasks and how the data is collected would help in understanding the evaluation setup better.

- The experimental setup seems weak and limited compared to the baseline (GPT-4V(ision) for Robotics).

- The authors could provide a video of the hardware demonstration to help understand the qualitative performance of the model and assess the effectiveness of the learned policy.

**Quality Of The Limitations Section:**

2

**Questions For Rebuttal:**

0) Address the weakness section in the reviews.

Additional questions:

1) I found the neuro-symbolic aspect of the work unclear. Although the term "neuro-symbolic" is used broadly, I am having difficulty relating this work to existing neuro-symbolic studies in the literature. It would be helpful if the authors could clarify why they consider this work specifically neuro-symbolic and clearly delineate what elements of their approach are neural versus symbolic. This discussion would enhance understanding and situate the work within the broader context of neuro-symbolic research.

2) In Figure 4(c), the "cube" seems to be a typo. It should be "bottle" instead.

3) what happens when multiple instances of an object are present in the scene? How the feature mapping will be done?

4) Are you considering releasing the code?

**Robotics Focus:**

4

**Summary Of Paper:**

The work tackles the challenge of learning a policy for a task by passively observing a human performing it. This problem is difficult because understanding human actions requires joint reasoning about the visual and language cues in the demonstration. Additionally, the learned policy should generalize to similar environments and objects. The proposed approach combines the visuo-lingual reasoning capabilities of vision-language models with task-specific neural networks and other hand-engineered components to learn a policy from human demonstrations and transfer it to new scenes and environments. The authors first preprocess the demonstration into a form suitable for the vision-language model to output the policy as Python code composed of high-level APIs. The evaluation is conducted in real-world manipulation tasks using the Stretch RE2 robot.

**Summary Of Recommendation:**

The paper proposes an interesting method for using vision-language models (VLMs) to learn a policy from human demonstration. In particular, it makes progress toward learning long-horizon policies that require concepts like loops and conditionals. This work could inspire similar research in the future. While the evaluation setting is limited compared to baseline, the inclusion of hardware experiments is indeed an advantage. Overall, I feel the strengths outweigh the weaknesses. I recommend that the paper be accepted, provided the authors address the concerns raised in the review during the rebuttal

---

### Author Rebuttal · Authors · 2024-08-14

Thanks again to all reviewers for taking the time and effort to help improve the paper.  We have uploaded a new manuscript version revised according to the reviewer comments and suggestions. The key changes include:
- Added implementation details for all underlying APIs, including expanded details about the scene transfer process. [Section A in appendix]
- Prompts used for program synthesis added [Figure 8 in appendix]
- Added figure illustrating correspondence matching for scene transfer with unseen objects, unseen environments, and both [Figure 5 in appendix]
- Added additional data collection details, experimental setup, and task descriptions for all tasks [Section B in appendix]
- Failure analysis and failure distribution figure added [Section C in appendix]
- Revised introduction to more explicitly define the context around the term neuro-symbolic.
- Revised limitations section to include more explicit and detailed discussion of assumptions made about the demonstration
- Revised limitations section to include discussion about resolving ambiguities between the visual and language components of the demonstration
- Updated related work section to include Dobb-E (Shafiullah et al 2023) and OK-Robot (Liu et al 2024)

---

### Decision · Program_Chairs · 2024-09-04

**Decision:**

Accept

**Comment:**

This paper introduces a programming by demonstration approach to generate robot programs, relying on a base set of foundation models to generate policies (program synthesis) for new tasks.

Strengths:
* Well written, method attempts to solve a challenging task, a good systems demo about an approach to integrate and generate programs by composing a set of lower level api calls provided by various base models and tools.
* A nice real-world demo

Weaknesses:
* There is a lack of clarity around the specific engineering implementations required to implement this system (what are the prompts, what information is required, what data is collected and what form etc. how were the low level primitives and controllers designed and implemented)
* There is insufficient detail around the tasks used and experimental details.
* While the method does produce a general program to solve complex tasks, this rules based approach requires all existing controllers and modules to be well engineered and functional
* The method does not seem to scale to handle faults or uncertainties, and scalability to longer horizon tasks seems limited
* The scientific contribution here is limited, particularly when considered with regard to prior work in program synthesis.

Reviewers provide a number of questions to be addressed in the rebuttal, in particular around missing information required for reproducibility and greater clarity around the novelty of the contribution.

Post rebuttal:
- The authors added additional detail around apis and prompts and experimental settings, along with greater discussion of limitations and failure analysis, and included more discussion of related work, addressing the bulk of the remaining weaknesses with this paper.